# The Concept of Family Farming in the Portuguese Political Discourse

**Isabel Dinis** [1,2]

[1]  Instituto Politécnico de Coimbra, 3045-601 Coimbra, Portugal
[2]  CERNAS - Research Centre for Natural Resources, Environment and Society,
    3045-601 Coimbra, Portugal; idinis@esac.pt

**Abstract:** Although several countries have outlined national and multi-criteria definitions, family farming is not well defined in most countries including Portugal, making it difficult to assess its real importance as well as the reasons underlying the design and the success/failure of particular policies. The main purpose of this study is to investigate the framing of family farming in the Portuguese political discourse by applying content analysis to a range of national policies and planning documents. The results show little reference to family farming in political documents and a conceptualization of family farming made in antagonism to professional or entrepreneurial farmers.

**Keywords:** family farming; content analysis; agricultural policy; rural

## 1. Introduction

Following the adhesion of Portugal to the European Economic Community (EEC) in January 1986, the Common Agricultural Policy (CAP) was adopted by the country, which led to profound socio-economic changes in Portuguese agriculture. Among other things, a large amount of structural funds was made available to support the modernization of agriculture, to promote structural adjustment and finally to transform small family farmers into entrepreneurs.

Since its beginning, CAP was designed and developed to support family farmers, the dominant typology in most Member States. As pointed out by Davidova and Thomson (2014), the government created an infrastructure that allowed small family farms to capture organisational-scale effects, without losing their specific features. However, the way subsidies were distributed, first based on output and later on area has generated controversy within the scientific and political communities. Some authors used the expression unequal distribution stating that subsidies have increasingly favoured larger farms rather than smaller ones (Beluhova-Uzunova et al. 2017; Burny and Gavira 2015; Hennessy 2014; Severini and Tantari 2013; Van der Ploeg 2016). Furthermore, the aims of the CAP have not always been clear, with conflicting discourses varying from productivism and neoliberalism to multi-functionalism, generating doubts about what was expected from farmers.

Nowadays, the role of family farms for smart, sustainable and inclusive growth is acknowledged at the European level (Davidova and Thomson 2014; Van der Ploeg 2016) and worldwide. Besides its role in food security, family farming is regarded as crucial for rural economies, because it generates income and employment, maintains the social vitality of the countryside and sustains rural landscapes and biodiversity. The recognition of the multiple roles played by family farming and the need to help family farmers become a more central focus of policy interests led the United Nations (UN) to declare 2014 as the UN International Year of Family Farming (IYFF). FAO (2014, p. 1) stressed the 'essential contribution of family farmers to food security, community wellbeing, the economy, conservation and global farm biodiversity, sustainable use of natural resources and climate resilience', a view widely shared by other international organizations. More recently, the UN

proclaimed 2019–2028 as the UN Decade of Family Farming, focusing on the design and implementation of collective economic, environmental and social policies in order to strengthen the position of family farming worldwide (FAO and IFAD 2019). Although not exactly focusing on family farming it is also important to highlight the adoption of the UN Declaration on the Rights of Peasants and Other People Working in Rural Areas by the UN General Assembly in December 2018, aiming to protect the rights of rural workers, including fishermen, nomads, indigenous peoples, pastoralists and other agricultural workers, such as peasants, which are defined in the declaration as 'any person who engages in small-scale agricultural production for subsistence and/or for the market, and who relies significantly, though not necessarily exclusively, on family or household labour and other non-monetized ways of organizing labour, and who has a special dependency on and attachment to the land' (UN 2018).

Following the declaration of the IYFF, the European Commission (EC) promoted a conference on the subject 'Family farming: A dialogue towards more sustainable and resilient farming in Europe and the world', preceded by a public consultation about the role of family farming, key challenges and priorities for the future. Throughout 2014 many family farming events were organized in many EU Member States, including Portugal. The debate and reflection carried out in Portugal allowed more in-depth knowledge about small family farming and emphasized the need for a Family Farming Statute (FFS) which was recently published by the Portuguese government. Nevertheless, as in other countries, it is still unclear what the perception of the Portuguese policy makers is towards the very concept of family farming and how they really address it in the design of sectorial policy. Furthermore, analyses of the discourse on family farming as developed by politicians, is of interest not only because it may explain the design of the agriculture policies, whether general or specifically targeted, but also their success/failure.

The aim of the present study is to understand: (i) How family farming has evolved in Portugal after adhesion to the EU; (ii) how family farming has been framed in Portuguese policy strategies, and how the framing has changed over time; (iii) if the perception of the Portuguese public administration has followed the evolution of the concept of family farming in the literature.

To answer the first question, statistical data, available in *Statistics Portugal*, was collected and analysed, with special emphasis on the Census of Agriculture. The second question will be addressed through the analysis of 28 public policy documents. Responding to the third question involves a conceptual analysis of the empirical findings in the context of the body of literature on family farming.

The paper is comprised of five sections. In the first section a review of the theoretical framework is provided, particularly regarding literature on the resilience of family farming and the discussion over the concept itself. In the second, the recent evolution of Portuguese family farming is presented. In the third section, the methodological approach used in the empirical investigation is explained. The following section provides the analysis of the results. Finally, conclusions, study limitations and opportunities for further research are highlighted.

## 2. Frame Analytic Approach

Although the end of family farming was predicted more than 150 years ago by Karl Marx (1867), family farms still represent more than 90 percent of the world's farms and about 75 percent of the world's agricultural land (Lowder et al. 2016). Following the orthodox Marxist tradition, both neo-classical economists and classical sociologists have predicted the disappearance or marginalization of family farms by submission to capital forces (Davis 1980; De Janvry 1980; Marsden et al. 1989; Whatmore et al. 1987a, 1987b). That is, increased technical efficiency deriving from economies of scale as well as the integration of agriculture in wider circuits of industrial and finance capital will progressively eliminate the family farm, unable to compete with agribusiness. As Shucksmith and Rønningen (2011, p. 275) highlight, 'doctrines of high modernism and neoliberalism have emphasised the virtues of economic efficiency, economies of scale and specialisation, while calling for deregulation and a minimalist state'. Structural change, leading to larger and more specialised farms was not only seen as an inevitable, but also as a desirable outcome.

The resilience of family farms within a capitalist context has been theorised in various ways over the past two centuries, most notably by Brookfield (2008), Brookfield and Parsons (2007), Chayanov (1966), Calus and Van Huylenbroeck (2010), Friedmann (1978, 1980), Schmitt (1991) and Shanin (1971). The main reasons presented by the authors to explain this resilience rest upon the distinct rationality of the family farm and on the use of family labour. Family farms aim to satisfy the family's needs rather than to make a profit, having more flexibility in the allocation of net returns between production expansion, inputs acquisition and family consumption, allowing them to compete successfully with profit maximization-oriented farms (Calus and Van Huylenbroeck 2010; Chayanov 1966; Friedmann 1978; Van der Ploeg 2000). The use of family labour enables the adjustment of labour intensification and personal consumption to internal and external changes and reduces fixed costs, derived from wages and labour supervision (Binswanger and Rosenzweig 1986; Calus and Van Huylenbroeck 2010; Gray 1998; Davidova and Thomson 2014; Gorton and Davidova 2004; Hazell 2005; Woodhouse 2010). Also, some authors (e.g., Gasson and Errington 1993; Hazell et al. 2010; Masters et al. 2013; Van Vliet et al. 2015) argue that many tasks are more efficiently done by family members, who are motivated and understand the local environment. Equally important, land productivity of small-sized farms is greater than that of large farms (Collier and Dercon 2014; Li et al. 2013; Tomich 1995; Van Vliet et al. 2015; Woodhouse 2010) and therefore, the economies of scale are relatively small compared to the advantages of optimal use of farm household labour (Calus and Van Huylenbroeck 2010; Schmitt 1991).

From the late 1980s, several studies have discarded the long-held assumption that family farms must inevitably give way to industrial farms and changed the focus from the internal aspects of the family farm household to the active strategies developed by farmers to achieve their goals (Barbieri et al. 2008; Barbieri and Valdivia 2010; Davidova and Thomson 2014; Knutson et al. 1998; Moran et al. 1996; Van der Ploeg 1993, 2000, 2016; Weltin et al. 2017; Wilson 2008). Intensification and specialisation, diversification to agricultural and/or non-agricultural enterprises (e.g., value-added production and rural tourism) as well as pluri-activity and pluri-income, are identified as important adaptation strategies to cope with market pressures and changing political framework conditions, and to reduce economic risk.

Often, the debate on small farms efficiency is based on economic considerations (productivity, profitability) and overlooks the other dimensions/functions of family farming. However, the way family farms are understood has important policy implications. Perspectives that favour economic framing may result in policy interventions that push towards specialization and land consolidation and compromise the more holistic notion of the family farm. As pointed out by Schneider (2016), public policies are often sectoral and disarticulated, not considering the internal dynamics between farm and family and their relationships with the social environment and the territory. As the authors exemplify for Latin America but extendable to other places in the world, many policies support production only by increasing scale and are not always connected or integrated with policies designed to promote the access to markets, building of new sales channels, environmental preservation and keeping people in the rural space.

Although family farms have figured prominently in the discourse about rural development, the concept of family farming varies widely depending on country, context, author and political motivation (Garner and de la O Campos 2014; Graeub et al. 2016). In most definitions the role of family labour is a crucial aspect, although the specifics on the amount of family labour used vary in the literature (Garner and de la O Campos 2014). In more multifaceted conceptualisations, the dependency on family labour is downplayed and other features are underlined. Although sometimes criticized for being non-operational (Djurfeldt 1996; Hill 1993) and thus inadequate for comparisons over time and between different societies, the definition by Gasson and Errington (1993) is the most commonly cited according to Garner and de la O Campos (2014). Besides labour, the authors put the emphasis on family members as farm managers that provide capital, live on the farm and are related by kinship or marriage. Moreover, in family farms, business ownership and managerial control are transferred between generations. From a sociological perspective, as stressed by Calus and Van Huylenbroeck (2010) and Davidova and Thomson (2014), family farming is associated with family

values, such as solidarity, continuity and commitment, being more than merely a professional occupation. It reflects a lifestyle based on beliefs and traditions about living and working. On the whole, the definition of 'family farm' has been subjected to profound debate. In-depth literature revisions concerning the concepts and features of family farming can be found in Brookfield and Parsons (2007), Van der Ploeg (2016) and Van Vliet et al. (2015).

## 3. Family Farming in Portugal

In many areas of the globe the concept of family farming is an heir to expressions such as 'smallholding' or 'peasantry'. As stated by Schneider (2016), despite the statistical advantages of using the concept of small-scale production, this term has become questionable, since land size says very little about the conditions of production and reproduction of farmers and does not distinguish between size (quantity) and scale (quality).

In Portugal, family farming has also been historically identified with subsistence-oriented smallholdings (minifundio) mainly present in the North of the country in opposition to large farms relying on wage labour (latifundio) that dominate in the South. The rural regions of the North have always been related to societies strongly structured around family farming, while in the South the importance of family farming was neglected to some extent. However, as showed by Carmo (2010) even in the South, family farming has always been present, although with some important differences regarding land tenure. While in the North family farmers owned their own land, in the South family farms were established on rented land. Often, the landlords rented some of the less productive plots to their own workers that used family labour to cultivate them. Most of these plots were clearly insufficient to support the farmer's family, contributing to the development of part-time family strategies, with family members dividing their time between the family farm and the landlord's cereal fields. The progressive mechanization and modernization of agriculture in the South and the steady decrease in the agricultural population increased the weight of farms mainly relying on family labour. Therefore, in spite of the remaining differences in land structure and ownership between the zones of 'minifundio' and the areas of 'latifundio' the presence of family farming increased noticeably in the latter, blurring the division between the two regions regarding family farming analysis.

As pointed out by Graeub et al. (2016), several countries such as Argentina, Brazil, Chile, Mexico, Uruguay and the United States of America, have outlined national, multi-criteria definitions of family farms and have used those definitions to analyse their census data. The European Union, although stating that the concept of family farming covers various sociological and economic elements (EC 2017), has never defined the concept precisely, frequently using the legal status of the farm as the sole criteria to identify family farmers. In the Eurostat Farm Structure Survey, normally, the family farmer is the sole holder, often (but not always) registered for statistical and policy purposes as a farmer but not constituting a legal business entity (Davidova and Thomson 2014).

In Portugal, family farms are usually identified with sole holders operating with predominantly family labour—autonomous sole holder—a statistic category present in the Census of Agriculture since 1979. However, since in the Farm Structure Survey of 2016 this category is not present, the criterion of the Eurostat Farm Structure Survey (sole holder) will be adopted. The analysis of base agricultural statistics, made available by *Statistics Portugal* (www.ine.pt), regarding 1989, 1999, 2009 and 2016 showed that the number of sole holders in Portugal decreased in absolute value 31.2 percent, 28.0 percent and 9.0 percent in the periods of 1989–1999, 1999–2009 and 2009–2016, respectively (Table 1). In relative value the number of sole holders decreased from 99.1% of the total number of farms to 95%.

**Table 1.** Number of farms and utilised agricultural area (UAA) in Portugal.

| Years | Total Farms | | Sole Holders | | Sole Holders (%) | |
|---|---|---|---|---|---|---|
| | Number | UAA (ha) | Number | UAA (ha) | Number | UAA |
| 1989 | 550,879 | 3,879,579 | 546,069 | 3,252,619 | 99.1 | 83.8 |
| 1999 | 382,163 | 3,736,140 | 375,938 | 2,935,907 | 98.4 | 78.6 |
| 2009 | 278,114 | 3,542,305 | 270,507 | 2,370,995 | 97.3 | 66.9 |
| 2016 | 258,983 | 3,641,691 | 246,149 | 2,273,881 | 95.0 | 62.4 |

Source: Statistics Portugal

A less sharp trend occurred with the agricultural area used by the same farms, which decreased 9.7 percent, 19.2 and 4.1 percent in absolute value in the same periods, representing a decrease from 83.8% (1989) to 62.4% (2016) of the relative weight of the agricultural area used by sole farmers. This movement was accompanied by the abrupt drop in the familiar agricultural population, which decreased by 37.6 percent, 36.8 percent and 20.8 percent in the same periods, accounting nowadays for 6.1 percent of Portugal's resident population. Nevertheless, the weight of family farms in the total number of farms and agricultural area has been more or less stable since 1989, representing around 95 percent of the total farms and more than 60 percent of the agricultural area. This scenario is not very different from what has occurred in most EU countries. In fact, family farms dominate the structure of EU agriculture in terms of their numbers, their contribution to agricultural employment and, to a lesser degree, to the utilized agricultural area. From the 10.8 million farms that existed in the EU in 2013, a vast majority of these (96.6%) are sole holders that cultivate 67.2% of the agricultural area (Eurostat 2013).

The analyses also showed an increase in the diversification of gainful activities and sources of income within the farm households, accentuating earlier tendencies described by Baptista (1994) and Carmo (2010). In fact, farm households with no 'other gainful activities' or pensions decreased from 11.5 percent of the total number of family farms in 1989 to 5.8 percent in 2016. The weight of retirement pensions in household income is very high, corresponding to the advanced age of the agricultural population. From the total family members carrying out agricultural activities on farms, only 13.9 percent worked full time in 2016 (16.3 percent in 1989). Almost one-fourth of sole holders (23.6 percent) reported having a gainful activity outside agriculture. According to Statistics Portugal (INE, 2017a), this was more prevalent among younger holders. 62.2 percent of the holders aged less than 40 carried out activities complementary to agriculture, while for those older the importance of other activities was residual (4.6 percent). Regarding the weight of CAP subsidies in income, the statistical data show that 40 percent of sole holders did not qualify for the payment of subsidies (direct payments and/or rural development measures).

For the remaining 60 percent, subsidies played an important role with one-third claiming that subsidies accounted for 25 to 75 percent of their income (INE 2017a).

## 4. Material and Methods

In order to examine the evolution of the concept of family farming in the public discourse in Portugal, the content of a range of public textual data published since the adhesion of Portugal to the EEC, comprising 12 government programmes (GP), 3 rural development programmes (RDP), 13 statistical analysis documents and the Family Farming Statute (Decree-law 64/2018 from 7th August) were analysed (Table 2). Textual data was gathered from the public online archives of the Portuguese Government, the Ministry of Agriculture and *Statistics Portugal*.

To understand how family farming is defined in the political discourse, GP are essential because they contain the main policy orientations of the Government, as well as the measures to be adopted or proposed in the various areas of government activity, including agriculture and rural development. On the other hand, RDP operationalize the political options of each EU Member State. They define rural development priorities and targets and set forth the measures and funding to achieve these targets. RDP are relatively new in the EU and were created following the Agenda 2000 CAP Reform. Since then, Member States have implemented RDP regarding three programming periods (2000–2006, 2007–2013 and 2014–2020).

Statistical analysis documents are not exactly policy documents. However, census of agriculture and farm structure surveys, present diagnosis of the agricultural situation used for monitoring trends and modelling policy proposals. Therefore, the way data is interpreted and communicated by national statistical agencies may have important implications on the political discourse. The FFS is a recent piece of legislation that, beside the government view, encompasses the ideas of the Inter-Ministerial Commission which wrote the initial proposal and the views of citizens and stakeholders

gathered through public consultation. It is a particularly important document to analyse because it explicitly focuses on family farming.

To examine the texts, content analysis was used. This method is widely used in research in the social sciences and allied disciplines. A comprehensive overview and detailed description of some important variations within the method are presented in Drisko and Maschi (2016). As highlighted by Heslinga et al. (2018) content analysis has high reliability and validity in the analysis of historical documents and changes in policy because the analysis can be done for any time period, even if participants in the events are unavailable or deceased, and because key informant's perceptions and reflections are likely to change over time. Several authors have been using content analysis to examine policy documents on different subjects, such as energy (Brondi et al. 2014; Kivimaa and Mickwitz 2011; Zhang et al. 2012), the environment (Kalaba et al. 2014; Erol and Yıldırım 2017), tourism and landscape (Heslinga et al. 2018), organic farming (Seufert et al. 2017) and sport promotion (Christiansen et al. 2014).

**Table 2.** Sources of data.

| Constitutional Government Programmes (GP) and Laws | Statistical Analysis Documents | Rural Development Programmes |
|---|---|---|
| X GP, 1985 <br> XI GP, 1987 <br> XII GP, 1991 <br> XIII GP, 1995 <br> XIV GP, 1999 <br> XV GP, 2002 <br> XVI GP, 2004 <br> XVII GP, 2005 <br> XVII GP, 2009 <br> XIX GP, 2011 <br> XX GP, 2015 <br> XXI GP, 2015 <br> FFS—Decree-law 64/2018 | Census of Agriculture 1999: Main results (INE 2001) <br> Census of Agriculture 1999: First results (Press release) (INE 2000) <br> Census of Agriculture 2009: Main results analysis (INE 2011a) <br> Census of Agriculture 2009: Preliminary data (Press release) (INE 2010) <br> Census of Agriculture 2009: Final data (Press release) (INE 2011b) <br> Farm Structure Survey, 1993, 1995, 1997, 2005, 2013, 2016 (INE 1995, 1996, 1999, 2006a, 2014a, 2017a) <br> Farm Structure Survey 2005 (Press Release) (INE 2006b) <br> Farm structure survey 2013 (Press release) (INE 2014b) <br> Farm structure survey 2016(Press release) (INE 2017b) | AGRO—Portuguese Rural Development Operational Programme 2000–2006 (MADRP 2002) <br> PRODER—Portuguese Rural Development Programme 2007–2013 (MAMAOT 2012) <br> PDR2020—Portuguese Rural Development Programme 2014–2020 (GPP 2014) |

To carry out this study, the content of each of the 29 documents was imported into the NVivo 11 Pro software for qualitative analysis, which was used to manage the data during analysis. As pointed out by Chambers et al. (2007), the main advantages of using computer packages is that text searches can be easily carried out, related themes and categories can be merged, and overlap between themes can be readily identified.

In this case, the general search for the expression 'family farm' and other derived or related expressions (e.g., 'family farming', 'family farms', household)[1] allowed the identification of the number of times the concept was mentioned in each document. The references to family farming were then contextualised using the software option 'coding context', which enables one to code the paragraphs surrounding the searched expression. This option allowed the identification of the context in which the expressions were referenced, rather than simply ascertaining the number of times they appeared in the documents. The empirical analysis continued by recoding the original coded paragraphs in two categories: non-conceptual and conceptual. The paragraphs were coded as non-conceptual if the references to family or household were disconnected from any qualifying attribute and were coded as conceptual whenever it was possible to detect an attempt to specify particular features of family farming. Only paragraphs coded as conceptual were considered for further analysis. Using previous literature review, a set of four deductive sub-codes was constructed

---

[1]   Since all the documents were in Portuguese language, the search was, in fact, made for the correspondent Portuguese words.

to contemplate several possible conceptualization paths (Features, Dichotomy, Strategies and Functions). The sub-code Features were applied when the conceptualization of family farm was based on family farm internal characteristics, such as size, type of labour, type of crop, yield and self-consumption level. Dichotomy was used to code paragraphs in which family farming was presented as a reality opposed to the industrial farming. Strategies coded paragraphs highlighting the differentiated strategies developed by family farms to achieve their goals, including intensification and specialisation, diversification to agricultural and/or non-agricultural enterprises, pluri-activity and pluri-income. Lastly, function was applied to paragraphs that defined family farming on a functional basis, highlighting the roles of family farming at the production, social and environmental levels.

Although content analysis is a well-established method in the social sciences, it may be reductive, particularly when dealing with complex texts (Dixon-Woods et al. 2005; Snilstveit et al. 2012). Abstraction of content from its context, such as taking a word or phrase in isolation from other parts of the text, may result in loss of meaning (Insch et al. 1997). In addition, content analysis may overlook what is not explicitly written in the texts and hinder the understanding of what is driving and shaping particular narratives. Although the texts analyzed in the present study are rather straightforward, at the end of the process, each document was read several times to ensure that the meaning of the conceptual paragraphs was consistent with the overall idea presented in the document.

## 5. Results and Discussion

The search for the expression 'family farm' and other derived or related words registered 463 occurrences (4 in GP; 30 in RDP, 408 in statistical analysis documents and 21 in FFS). However, after coding context and further reading, it was possible to recode most of the paragraphs as non-conceptual and drop them from further analysis. Particularly in statistical analysis documents, several references to family farming were headings, table captions or mere presentations of results without any interpretation purposes. Only 18 paragraphs (4 in GP, 7 in statistical analysis documents, 3 in RDP and 4 in FFS) were coded as conceptual. The narrative summaries that reveal the key issues presented in the conceptualization of family farming in the target public documents are presented in Tables 3–6.

**Table 3.** Conceptual references to family farming in the Portuguese Government programs.

| Government | Context | Sub-Code |
|---|---|---|
| X CGP, 1985 | The establishment of family farming companies will be encouraged. | |
| XII CGP, 1991 | (…) to preserve the family farm model not only with economic capabilities but also social, cultural and nature protection, to meet multifunctionality requirements preconized in the principles of the Common Agricultural Policy reform. | Function/Strategies |
| XIV CGP, 1999 | Promote an integrated vision of rural development with a view to sustainability and social and territorial equity, (...) specifically supporting small family farming and encouraging multifunctionality of the farm. | Function/Strategies |
| | In addition to the incentive system aimed at boosting competitiveness, there will be, in the case of agriculture and rural development, a specific regime for small family farming (…). | Function |

Out of the 12 GPs presented by the Portuguese Governments since 1985, only 3 explicitly addressed the issue of the family farm (Table 3). Not even the present government that took office in the year immediately following the IYFF provide any guidance or political measures specifically regarding family farming. Small farming is mentioned several times but with no clear relation to family farming.

In the 3 GPs that mention family farming, the need to support this type of farming is stressed. Although with no evident conceptualization basis, the function of family farming in rural sustainable development is explicitly recognised, particularly in the 1991 and 1999 GPs. Multi-functionality is also mentioned, highlighting one of the strategies developed by family farmers in order to cope with the changing forces in rural areas. This vision is probably linked to the CAP trends at the time. At this period a deep restructuring of the CAP was taking place, with the withdrawal of several price support mechanisms and incentives for cultivated area reduction. Rural areas were starting to be seen not only as places where people lived and worked, but also as territories with vital functions for society as a whole, by providing ecological equilibrium and a refuge for relaxation and leisure (ECC 1988). Diversification was then presented as a strong rural development strategy and multi-functionality as a way to improve farmers' income.

A vision of family farming, based on its functions and strategies, is adopted in the main policy documents, up to 2009. In PRODER, family farming is implicitly associated with the strengthening of the rural economy and multi-income (Table 4).

**Table 4.** Conceptual references to family farming in the Portuguese Rural Development Programmes.

| Program | Approach/Context | Sub-Code |
|---|---|---|
| PRODER (2007–2013) | Family farming is implicitly associated with multi-income, population stabilization, land use and strengthening of the rural economy. | Function/Strategies |
| PDR 2020 (2014–2020) | Family farming is explicitly associated with:<br>– Small area and small economic size (the terms family farm and small farm are used interchangeably)<br>– Low specialization<br>– Pluriactivity and pluri-income<br>– Low opportunity costs. | Function/Features/Strategy |
| | Family farming is conceptualized as opposed to professional farming: | Dichotomy |
| | <u>Family farming</u><br>**Features:**<br>- Uses mainly family labour;<br>- Small and very small economic dimension;<br>- Represents the majority of farmers;<br>- Less important in terms of production value and agricultural area.<br>**Functions:**<br>Essential for preserving the environment and managing of natural resources; for the human and economic occupancy of rural areas and for social inclusion, also representing an important part of the supply of agricultural goods.<br>Social response or poverty alleviation for many people, often elderly and with low levels of education. | <u>Professional farming</u><br>**Features:**<br>- Uses hired labour in a greater proportion;<br>- Large and medium economic dimension;<br>- More specialized;<br>- Occupies the bulk of the agricultural area;<br>- Represents a smaller number of farmers;<br>- Responsible for most of the production.<br>**Functions:**<br>Important role in terms of competitiveness of the Portuguese economy, with a productivity similar to the rest of the economy. |

Since 2009, not only in statistical analysis documents (Table 5) but also in development programmes, the concept of family farming is based on a binary division of producers. On one hand, there are a relatively small number of professional or entrepreneurial farmers, driven by individual economic goals reduced to profit maximization agents that make a major contribution to food production and economic competitiveness but have no role in rural vitality or environmental wellbeing. On the other hand, there are a large number of small family farmers, barely competitive, but essential for sustainable rural development. It is clear, particularly in the 2013 farm structure survey and PDR 2020, that a hierarchical significance is ascribed to this dichotomy. The corporatisation of agriculture is seen as a virtuous path because it contributes to an increase in efficiency in agriculture, due to the adoption of more professional management processes and economies of scale, allowing the agricultural sector to reach a productivity level similar to the rest of the economy. This kind of conceptualization conveys the message that efficiency in food production is not expected from family farming and that environmental and social functions (biodiversity, landscape, water management, food security, animal welfare, etc.) are not in the scope of entrepreneurial farmers.

**Table 5.** References to family farming in *Statistics Portugal* publications.

| Publications | Approach/Context | Sub-Code |
|---|---|---|
| Farm structure survey 1995 | Agriculture is based on the concept of family farming, in which labour is mainly ensured by the producer and members of his/her household. | Features |
| | Our agriculture relies mainly on family labour, with a strong traditional structure. | Features |
| Farm structure survey 2005 | A family farm is a farm in which labour is supplied by the producer and the members of his/her family, who do not receive a salary, represent about 75% or more of all the labour used in the farm. | Features |
| Census of agriculture 2009 | The high representativeness of family farming coexists with the reality of entrepreneurial agriculture, formed by agricultural corporations. | Dichotomy |
| Farm structure survey 2013 (including Press release) | The reality of agricultural enterprises is very different from that of more familiar farms (...) | Dichotomy |
| | The corporatisation of agriculture expressed by the growing number of agricultural enterprises has contributed to increase the efficiency of the sector because of the adoption of more professional management processes and economies of scale. | Dichotomy |
| Farm structure survey 2016 (including Press release) | The high representativeness of family farming formed by small holdings, thus coexisted with large-scale and entrepreneurial agriculture, mostly composed of agricultural enterprises that although accounting for only 4.4% of holdings in 2016, managed almost one-third of the UAA and produced 44.6% of the livestock. | Features/ Dichotomy |

The conceptualization of family farming based on its features is present at two levels. In the first, the only feature used to describe family farming is the dominance of family labour. This approach is mainly present in statistical analysis documents before 2009, with the purpose of enabling data analysis. At another level, especially visible in PDR 2020, several features of family farming are pointed out (family labour, small area and small economic size, low opportunity costs). However,

the purpose is not to establish a multi-criteria definition of family farming but rather to emphasize its opposition to professional farming. In this sense, being a professional farmer means having a large area and a large economic dimension, being specialized, efficient and competitive in food production. Excluded from this view of the agriculture profession, are not only small farmers, but also all the farmers that choose polyculture or on-farm business diversification as farm strategies.

FFS is the most recent text analysed in the research. In this document, the role of family farming at the economic, social and environmental levels is stressed. However, the functional aspect of the family farming concept is not the only one that is mentioned. The definition of a formal concept is mandatory since the document focused on positive discriminatory policies favouring family farmers. In the operational definition of the concept, small economic size and family labour are the selected features (Table 6).

**Table 6.** Conceptual references to family farming in laws.

| Law | Context | Sub-Code |
| --- | --- | --- |
| FFS (2018) | Several references to the role of family farms in local economies (production, consumption and employment) in public goods and services provision (biodiversity and environment preservation), in food losses and wastage minimization and in preventing interior rural areas depopulation. | Function |
| | Family farming definition mainly based in total family collectable income (no more than 2500,000 euros per year), amount of direct payments (no more than 500,000 euros per year) and the use of family labour (family labour must represent at least 50% of total labour) | Features |

Analysis shows that the perception of the Portuguese public administration has hardly followed the evolution of the concept of family farming in the literature. In several Portuguese policy documents, including the recent FFS, family farming is presented as synonymous to small scale and poverty, in opposition to markets and technology; although, as stated by Schneider (2016), the current discussions on family farming are overcoming this bias. As the author highlights, the social and economic reproduction of family farms is no longer restricted to the small rural communities or to isolated villages. Interaction with the broader society and markets allows their social reproduction in different societies and economies, including the capitalist mode of production.

The dichotomous perspective, connecting family farming with features like continuity, risk avoidance and small size and linking entrepreneurial farming with specialization, scale enlargement, profit maximization and risk taking are shared by several authors (Austin et al. 1996; Davis-Brown and Salamon 1987; Marsden 1984; Van der Ploeg 2003, Van der Ploeg et al. 2009). However, more recently, some literature has pointed out that family farming in the EU covers a wide range of farm types and sizes.

The recent literature also shows that even at the functional level, the dichotomy between family farms and industrial farms is not so evident. The fact that some family farms have focused on commercial farm business operations, and that some large-scale non-family farms are starting to be involved in multi-functionality and ecological entrepreneurship, makes these two farm categories much more compatible with each other (Davidova and Thomson 2014; Niska et al. 2012; Renting et al. 2008).

In Portuguese political documents, the agriculture profession is often, explicitly or implicitly, identified with large farms, large economic dimension, specialization, efficiency and competitiveness, excluding diversification as a professional strategy. However, as pointed out by Renting et al. (2008), the motivations of both family farmers and entrepreneurial farmers are changing with consequences for professional identities, extending the boundaries beyond what is traditionally known as 'agriculture'. Furthermore, as the authors claim, changing occupational identities may result in additional difficulties to agricultural policy makers, 'with a growing discrepancy between

regulations that define the formal status of agricultural activities and the "real" world of activities' (Renting et al. 2008, p. 17).

## 6. Conclusions

The aim of this study was to examine the Portuguese Government's narrative on family farming from Portugal's adhesion to the EEC to the present. Family farming is often described as the predominant form of agriculture in Portugal, although, as in most countries, family farmers are not a well-defined group.

One of the main conclusions of the study is that there is little reference to family farming in political strategic documents related to the agricultural sector. The results also show that the representations suggested by politicians are less centred on specific features of family farming and more on the contrast between family and entrepreneurial or professional farming. In the Portuguese political discourse, family farmers and entrepreneurial farmers have been commonly perceived as contradictory farmer categories. Like in the neoliberal discourse, dominant by the end of the past century, structural change, leading to larger and more specialised farms is seen as desirable. The farming profession is identified with models exclusively oriented to productivity, excluding activities related to their environmental and social functions. The importance of farm multi-functionality is recognized, but only for family farming.

The political discourse has not been able to integrate the diversity and evolution of family farming, neither the compatibilities between the two classical types of farms, with implications for the ability to incorporate the increasing complexity of farming in agriculture policy. Taking into account all types of farms, whatever their objectives and articulation with marketable goods and services, will allow the conception of more suitable state intervention and support measures for the Portuguese reality, thus avoiding the exclusion of 40 per cent of family farms from the agriculture policy benefits.

This article was written as an initial effort to present the Portuguese picture concerning public discourse on family farming. It should be noted that the Portuguese discourse is probably not independent of the European discourse. However, the paper does not investigate the relation between the trends in the Portuguese and the European discourses and policies concerning family farming. Future research centered on the comparison between European planning and policy documents and the Portuguese documents will build on this paper's findings and fill some of the gaps in the present analysis. Further research is also needed to understand how other stakeholders, such as farmer's associations conceptualize family farming in order to identify different discourses at different levels and reach a common multi-criteria definition of family farming that can be used for statistical and analytical purposes as well as for the design of more effective agricultural policies.

**Funding:**

This research received no external funding.

**Conflicts of Interest**:

The author declares no conflicts of interest.

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
