# Peer review of "The Concept of Family Farming in the Portuguese Political Discourse"

_socsci, doi:10.3390/socsci8070213_

Round 1

Reviewer 1 Report

The article reports an content analysis of Portuguese policy and other documents to analyze how family farming has been framed in these texts. The originality of the article lies in the method used. 

However, the difficulty with the article is that it brings besides the method hardly something new:  what does it brings as knowledge besides that family farming is rarely used as concept in policy documents and that it is opposed to ‘large industrial farming’. This is to my opinion known already and is not different for Portugal as opposed to other countries. So the main question is what can we do with the analysis and conclusions from this article !

Further, the authors give in the introduction three objectives but as far as I can see only the first two are really tackled in the article? The third one: If the perception of the Portuguese public administration has followed the evolution of the concept in family farming literature is hardly analysed (only indirectly). 

Further the statistical analysis of the family farming situation in Europe and Portugal reamins weak. For EU no figures are given and het figures for Portugal in decrease of family farms are misleading because they only report the decrease in general in the number of farms but do not say anything on whether family farming less or more important.

Finally, I also find that the authors start from a predefined thinking as whether family farming is better than Industrial farming (the old model). This shows that although the authors criticise thati n policy documents no real defintion is given of family farming, they als have a perceived defintion in mind (the Schumpeter: 'small is beuatiful' way of thinking). I think that much more emphasis can be laid on the fact that the concept of family farming has not necessarily something to do with small or large but with ownership: who is het main owner and decision maker in the holding (there fore family holding is to my opinion a better word !) because also larger and so called Industrial farms can be family businesses (and there are many examples of that !)

Therefore I would ask the authors to try to embed their original analysis method in a wider and more objective analysis framework: what do they really want to know and what is het pupose of the anaysis and to start with an theoretical analysis why it is important to make a distinction in policy papers and decsion making between family and non-family farming? Maybe the conclusion is that it is not so important (first because only am inority of farmers are not belonging to het category and so there is no purpose to exclude this last category, or maybe because het concept as such doesnot help to reach policy goals). In any case the conclusions about the relevance of the results should be made stronger!

Some detailed remarks:

-          P1: l2 2 Portugal was forced: It was a free decision to joint he EU so wording should be different. Portugal accepted to … (or something similar)

-          L23: unequal distribution<. This is an opinion, not a fact. Support is divided on the basis of objective criteria ! Write the article in a scientific not tendinous way !

-          L77: give some figures for EU as the article is about EU farming (and later about Portugal)

-          L91 Calus and van Huylenbroeck must be Calus and Van Huylenbroeck

-          L96 and L99 , L106 Idem (Calus and Huylenbroeck)

-          L147: give the exact numbers and percentages, not only the decreases (as of course also the number of farms has decreased !) (see line 154): so this is a bit odd !

-          L278 : preponderance: better dominance ?

-          L317: I have a difficulty with this hypothesis and have the impression that family farm is often used as a synonym for small farming, which leads to the confusion (not only in Portugal). Of course a large ‘industrial type of farm’ can also be a family owned farm. I think this needs more discussion

Author Response

Thank you very much for your valuable review. Please see attachment.

Reviewer 2 Report

 This manuscript offers a competent and original empirical analysis of the concept of family farming within the Portuguese policy framework post 1986. The paper reads well and is clear and insightful. This reviewer identified some gaps for the author(s)’ consideration:

·       Review of scholarly debate on family farming (section 2): the distinction between family farming as a political and social construct and family farming as an economic unit is worth greater emphasis. The debate on small farms efficiency is based on economic considerations (productivity, profitability) and overlooks the other dimensions/functions of family farming mentioned in the paper. How family farms and understood has important policy implications and this worth a reflection. For example, perspectives that favour the economic framing may result in policy interventions that push towards specialisation and land consolidation and compromise the more holistic notion of the family farm (as seems to be happening in Portugal). The Brazilian experience with public policies directed to family farms and their long term implications (potentially undermining the more holistic perspective and pushing farmers towards ‘conservative modernisation’) is worth considering here (see for example work by Sergio Schneider).

·       Overview of family farming in Portugal (section 3): there is no reflection in the paper on how the profile of family farms may vary within countries, across regions or across farming systems - one would expect variations between the minifundio vs latifundio agrarian structures found in the North vs South of the country. These are worth a comment and how other features besides size may also vary accordingly. Are national statistics and policy documents able to capture these nuances? If not, that would be a point worth making.

·       Methodology (section 4): the link between the four sub-codes used for content analysis and the literature and scholarly debates could have been explicitly outlined to offer a more elaborate and detailed specification of each of the sub-codes: for example - ‘features’ are presumably about size, use of labour, places of living and main sources of income (each of these links to particular contributions in the literature). Although content analysis is a good method for capturing dominant narratives, it has limitations in terms of interpreting their significance or where they come from (for example, politically, what is driving and shaping particular narratives?). A brief discussion of shortcomings of the method and how these may require additional research would have been appropriate.

·       Analysis (section 5): The analysis is clear but could have been more ambitious - one is left wanting to know more. Can there be a brief reflection on the historical trajectory of discourse, if a clear pattern can be found? What may have prompt the binary framing since 2009? How can historical patterns in Portuguese discourse on family farming be linked to (explained by) trends in European discourse on in the Portuguese domestic political landscape? If these cannot be immediately answered, it is worth expanding on further research needed to build on this paper’s findings and fill some of the gaps in the analysis.

Minor editorial points

·       Title of table 1 should perhaps be renamed as “Sources of data”

·       Editorial proof-reading necessary for typos, use of English language and punctuation.

Author Response

Thank you very much for your valuable review. Please see the attachment.

Round 2

Reviewer 1 Report

The article is much improved and my comments mostly incorporated; I only have two minor remarks

Line 23

However, the benefits were distributed in a highly unequal  way, increasingly favouring larger farms rather than smaller ones (Beluhova-Uzunova et al. 2017, Burny and Gavira 2015, Hennessy 2014, Severini and Tantari 2013, Van der Ploeg 2016).

My criticism was not that the authors cannot have an opinion but it should be written in a neutral way and make other opinions possible. Of course subsidies are distributed based on output (old system) or area (new system). This is an objective fact. But this is something different from the word ‘benefits’ which is tendinous. This is by the way proven by the fact that still majority of farms in Europe are family farms ! I would suggest to write e.g.:

However, subsidies were distributed based on output or area what makes that the distribution can be discussed; certain authors even use the term unequal distribution as according to them they increasingly favour larger farms rather than the small ones (………………………)

Line 180: be more correct and precise !

The analysis of base  agricultural statistics, made available by Statistics Portugal (www.ineNaN), regarding 1989, 1999, 2009 181 and 2016 showed that the number of family farmssole holdersin Portugal decreasedin absolute value 31.2 percent, 28.0 percent 182 and 9.0 percent in the periods of 1989-1999, 1999-2009 and 2009-2016, respectively (Table 1).In relative value the sole holders decreased from 99.1 % of the total number of farms to 95 %.

 (PS do the same with the subsequent sentences as also there there raemains confusion between the table and the percentages given !)

Author Response

Response to Reviewer 1 (questions in black and answers in blue)

All the comments were accepted and changes were made accordingly.

1)    Line 23

However, the benefits were distributed in a highly unequal  way, increasingly favouring larger farms rather than smaller ones (Beluhova-Uzunova et al. 2017, Burny and Gavira 2015, Hennessy 2014, Severini and Tantari 2013, Van der Ploeg 2016).

My criticism was not that the authors cannot have an opinion but it should be written in a neutral way and make other opinions possible. Of course subsidies are distributed based on output (old system) or area (new system). This is an objective fact. But this is something different from the word ‘benefits’ which is tendinous. This is by the way proven by the fact that still majority of farms in Europe are family farms ! I would suggest to write e.g.:

However, subsidies were distributed based on output or area what makes that the distribution can be discussed; certain authors even use the term unequal distribution as according to them they increasingly favour larger farms rather than the small ones (………………………)

Please see the new text (green text)  in lines 23-26.

2)    Line 180: be more correct and precise !

The analysis of base  agricultural statistics, made available by Statistics Portugal (www.ineNaN), regarding 1989, 1999, 2009 181 and 2016 showed that the number of family farms sole holders in Portugal decreased in absolute value 31.2 percent, 28.0 percent 182 and 9.0 percent in the periods of 1989-1999, 1999-2009 and 2009-2016, respectively (Table 1).In relative value the sole holders decreased from 99.1 % of the total number of farms to 95 %.

 (PS do the same with the subsequent sentences as also there there raemains confusion between the table and the percentages given !)

Please find the improved text between lines 183-191